# Urbanization strengthens vertical stratification of ant nutrient preferences in a temperate forest ecosystem

**Michelle Kirchner**[1,2,3]*, **Lucie Ciccone**[2,3], **Clyde Sorenson**[2], **Elsa Youngsteadt**[3,4]

**1** U.S. Department of Agriculture–Agricultural Research Service, Pollinating Insect–Biology, Management, Systematics Research Unit, Logan, Utah, United States of America, **2** Department of Entomology and Plant Pathology, North Carolina State University, Raleigh, North Carolina, United States of America, **3** Department of Applied Ecology, North Carolina State University, Raleigh, North Carolina, United States of America, **4** Center for Geospatial Analytics, North Carolina State University, Raleigh, North Carolina, United States of America

* michelle.kirchner4784@gmail.com

## Abstract

Resource and nutrient availability varies spatially and influences animal foraging patterns. Under the compensation hypothesis, animals should preferentially forage for the most limiting nutrient in the environment. Animal nutrient preferences have been well studied in the tropics, where terrestrial and arboreal fauna are clearly differentiated and limited by different nutrients. In temperate forests, vertical stratification of the fauna may be less pronounced and its role in foraging ecology is poorly understood. Here, we examine nutrient preference patterns over a vertical gradient in temperate forests and nearby urban centers in North Carolina, USA. Using a bait-choice experiment and novel bait design, we measured ant community nutrient preferences in the canopy and on the ground of 83 trees across 14 sites and assessed ant diversity and community composition. Ant diversity did not differ across the vertical strata or habitat types, but species turnover altered community composition to create four distinct assemblages. In temperate forests, ants did not prefer a particular nutrient in either stratum, likely due to extensive foraging across strata. In urban habitats, however, ant nutrient preferences matched patterns well known from tropical systems: arboreal animals preferred protein, while terrestrial animals preferred carbohydrates. Rather than stratum-specific nutrient limitations, however, we attribute the differences in urban preference patterns to changes in native species' foraging intensity and the addition of uniquely urban species with specific nutrient preferences. These results underscore the necessity of testing ecological hypotheses across biomes and suggest that urbanization may produce established ecological patterns via novel mechanisms.

## Introduction

Animals forage to meet their nutritional needs to survive and reproduce. Because nutrient availability varies across landscapes and trophic levels, animals must adjust their feeding behaviors to regulate nutrient intake toward an optimum level [1]. According to the compensation hypothesis, animals preferentially forage for the most limiting nutrient in the

**Data availability statement:** Data and code are available at the following link: https://doi.org/10.5061/dryad.cvdncjt9n.

**Funding:** MK was supported through a Research Assistantship with North Carolina State University Department of Entomology and Plant Pathology.

environment [2]. This "bottom-up" effect can influence foraging behavior at a range of spatial scales, with consequences for community composition, trophic webs, and species interactions [e.g., 3–7].

In forests, nutrient availability varies along a vertical gradient due to fundamental differences in the dominant food webs across strata [3]. Living vegetation in the forest canopy serves as the base of the arboreal "green food web," where C:N ratios can exceed 40:1 [8]. In contrast, the forest floor is fed by detritus of the "brown food web," and decomposition increases nitrogen levels, such that C:N ratios are 10:1 [9]. Following the compensation hypothesis, arboreal animals should preferentially forage for protein-rich foods to offset the relatively low nitrogen availability, but because nitrogen is plentiful in the "brown food web," animals in the leaf litter should prefer carbohydrate-rich foods [2,6]. Indeed, canopy-dwelling primates expend more relative foraging effort on high-nitrogen resources than expected based on their caloric value [10], and tropical ant communities prefer protein baits in the canopy and sugar baits on the ground [e.g., 2,3,5,but see 11]. While vertical stratification of animal nutrient preferences is relatively well studied in tropical forests, we know little about patterns in temperate forests [but see 12]. Even though the vertical gradient in relative carbon and nitrogen availability is similar in foliage and leaf litter across temperate and tropical broadleaf forests [13], nutrient availability at higher trophic levels may depend on site-level factors, such as local climate, physiography, biotic communities, and disturbance [8,14]. For example, nutrient cycling rates generally decrease with latitude, such that tropical forests exhibit higher nutrient turnover than temperate or boreal forests [15]. Additionally, plant diversity and resources are concentrated on the forest floor in temperate zones, while plants are more evenly vertically distributed in spatially complex tropical forests [16], which can have cascading trophic effects [17,18]. Given the climatic and structural differences between temperate and tropical forests that could result in distinct nutrient dynamics, it is unclear whether animal nutrient preference patterns known from tropical forests should persist in a temperate setting, leaving a key gap in our grasp of trophic ecology in temperate forests.

A clearer understanding of temperate forest ecology is critical at a time when temperate forests, already the most heavily fragmented forest biome globally [19], are expected to suffer significant losses from urbanization in the near future [20]. Urbanization impacts every level of the trophic web by altering resource quality and availability [21]. Although foliar and soil C:N ratios are similar between urban and rural forests [22,23], animals in cities also have access to anthropogenic food inputs in the form of food waste and rubbish [24]. These additional resources tend to be heavily biased toward carbohydrates, altering nutrient availability such that urban animals shift their dietary preferences and nutrient intake [24–26]. The addition of carbohydrate-rich anthropogenic resources on the urban ground may balance the high nitrogen content of the terrestrial "brown food web," but this possibility has not been tested in the context of the compensation hypothesis.

To address these knowledge gaps, we assessed ant community nutritional preferences on the ground and in the tree canopy of temperate deciduous forests and nearby urban areas. Ants (Hymenoptera: Formicidae) are an established model for nutritional ecology [e.g., 3,11], as they are abundant in both epigaeic and arboreal habitats and exhibit striking diversity across trophic levels [2,6]. We characterized ant diversity, community composition, and nutrient preference (protein vs. carbohydrate) in both habitats (forest and urban) and vertical strata (canopy and ground). We expected community composition in both habitats to follow established patterns in non-urban habitats, wherein temperate ants are more diverse on the ground than in the canopy [27]. We also expected non-urban sites to harbor more diversity than urban sites in both strata [e.g., 28]. We used these communities to test the following nutritional predictions:

1. Per the compensation hypothesis, ants foraging in the temperate forest canopy will prefer protein, while those on the temperate forest ground will prefer carbohydrates.

2. Urbanization will eliminate nutrient preferences on the ground, but nutrient preferences in the canopy will remain unchanged.

## Methods

### Study sites

Field work took place in the Piedmont ecoregion of central North Carolina (NC), USA, which falls under a temperate-group Köppen-Geiger classification (Cfa: humid subtropical) [29] with mean monthly temperatures ranging from -2.8°C in January to 31.7°C in July [30]. The Piedmont ecoregion of the southeastern United States covers ~ 166,000km$^2$ and contains major metropolitan areas, including Atlanta, Charlotte, and Raleigh-Durham. The region is under intense urbanization pressure, and developed land is expected to triple between 2009 and 2060 [31].

At seven forest and seven urban sites, we selected three healthy, mature trees from each of two common species: white oak (Fagaceae: *Quercus alba* L.) and red maple (Sapindaceae: *Acer rubrum* L.) (Fig 1a). Both species are host plants for honeydew-producing insects known to attract ants [32]. Forest sites were in protected, secondary mixed hardwood forests in state parks, game lands, or private property, and urban sites were in metropolitan areas of Raleigh and Durham on city-owned rights-of-way, which included street medians, residential lawns, parking lots, commercially maintained landscaping, and sidewalks. Sites were separated by at least 2km, and sampled trees within a site were at least 20m but no more than 1000m apart. Forest fragments ranged from 8ha to 100ha of continuous forest cover, and urban sites consisted of planted street trees (Fig 1a). All field work was conducted under North Carolina Division of Parks and Recreation Scientific Research and Collection Permit #2021_0490, North Carolina Wildlife Resources Commission Wildlife Collection License #21-SC01368, permission from the City of Raleigh Parks, Recreation and Cultural Resources Department, permission from the City of Durham General Services Department, or private owner permission. This study did not require ethical approval for animal care.

### Ant sampling and biomass

We sampled ants on one dry day per site between June 30 and August 9, 2021 using a novel bait station design (Fig 1b, S1 Text in S1 File). Each bait station held a pair of baits, and each bait contained ~ 6 cm$^3$ of either a carbohydrate-rich (strawberry preserves, Smucker's®) or protein-rich food (tuna, StarKist® chunk light in water, drained) (S1 Table in S1 File). At each tree, we placed four bait stations, two in the canopy (mean height: 14.6m, sd: 5.9m, range: 2.1–31.1m) (S3 Text, S1 Fig in S1 File) and two on the ground (5m N and S of the base of the tree) (Fig 1c) between 9:00 and 11:30am. After 4 and 8hr, we collected all ants on each bait into 70% ethanol. Baits were completely replaced after the 4h collection. Ants were identified to species using genus-specific keys [e.g., 33,34] (S2 Table in S1 File). Vouchers are deposited in the NC State University Insect Museum. To determine nutrient preference, we sorted all ants from each bait by species, dried (24h in a 55°C dry block followed by 48h in a 55°C drying oven) and weighed them to the nearest 0.01mg with an analytical balance (Mettler Toledo MS105DU, readable to 0.01mg) or microbalance (Mettler Toledo XP6, readable to 1μg). Ants of the same species from the same bait were dried and weighed together, and all baits were dried and weighed independently.

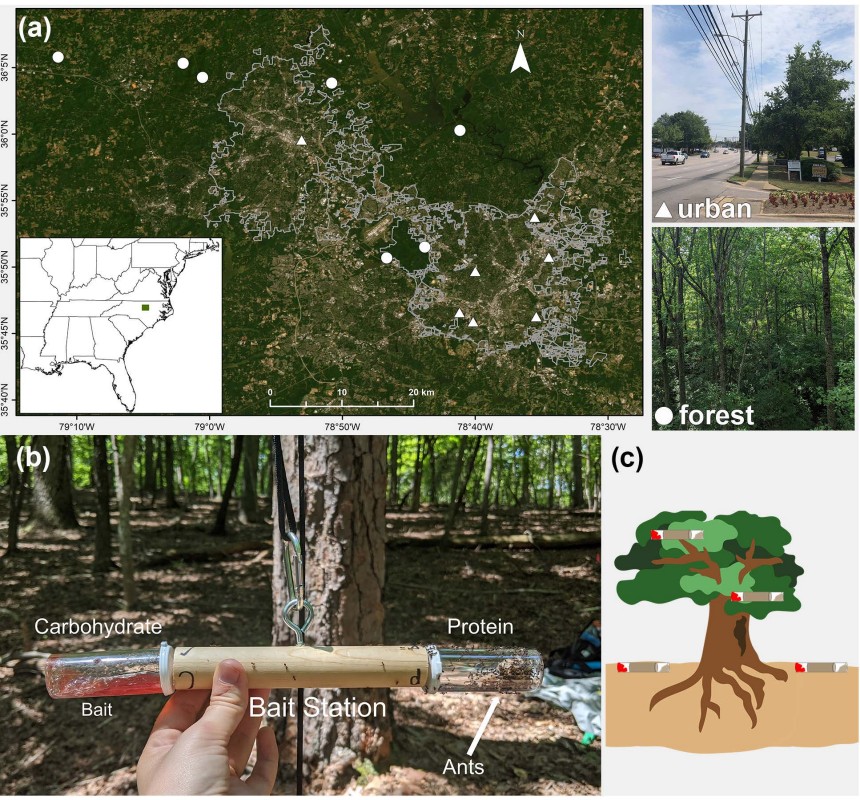

**Fig 1. Study design and methods.** (a) Map of 14 study sites in central North Carolina. Gray outline indicates Raleigh and Durham city limits. (b) Photograph of bait station design. (c) Illustration of bait station placement at each tree. Illustration credit: J Schlauch. Photo credit: M Kirchner (forest, bait), L Ciccone (urban). Map created using ArcGIS® software by ESRI. Displayed data were collected in the field by M Kirchner and L Ciccone. ArcGIS® and ArcMap™ are the intellectual property of Esri and are used herein under license. Copyright © Esri. All rights reserved. Basemap credits: World Imagery: NC CGIA, Earthstar Geographics.

## Data analysis

Data analyses were performed in *R* 4.2.2 and *RStudio* 2023.3.1.446 [35,36]. In all analyses, we considered each site as an independent replicate; non-independence of trees within sites, bait stations within trees, and baits within bait stations is represented by nested random effects, fixed nuisance variables, or pooling of data where appropriate.

**Diversity and community composition.** We pooled all baits within each combination of site and stratum (N = 28) and recorded each ant species' presence or absence. To assess differences in ant species diversity between strata (arboreal/terrestrial) and habitat types (forest/urban), we performed a sample-based rarefaction and extrapolation using *iNEXT* 3.0.0 [37,38]. Then, we removed singleton species to avoid undersampling bias and tested for changes in community composition in *vegan* 2.6-4 [39]. First, we conducted permutational multivariate analysis of variance (PERMANOVA) using Jaccard dissimilarity indices. We included the effects of stratum, habitat type, and their interaction, and an additive (not interactive) effect of site as predictors in "adonis2", performed 999 permutations for each test, and assessed the marginal effects of each term. We calculated pairwise comparisons between each level of the stratum*habitat interaction by iteratively passing the appropriate subset of the distance matrix to "adonis2", performing 10000 permutations of each test, and then, adjusted for multiple comparisons

using Benjamini & Hochberg's [40] method for controlling the false discovery rate (public communications, Maurice Codespoti Goodman, pairwise_permanova.R, accessed June 6, 2023, https://gist.github.com/mcgoodman/58c9d1257fd1625954a4ffa1c3301939). We also tested for multivariate homogeneity of dispersion around each group's median (PERMDISP2) with "betadisper". To visualize shifts in community composition, we performed non-metric multidimensional scaling (NMDS) on the species occurrence Jaccard distance matrix, ordinated on three dimensions using "ordiplot3d" in *vegan3d* 1.1-2 [41]. Finally, to clarify the role of species replacement versus loss in separating the canopy and ground communities in each habitat and the urban and forest communities in each stratum, we computed species turnover and nestedness between the canopy and ground for each site and between the habitats for each stratum at each site using "beta. pair" in *betapart* 1.6 [42].

**Nutritional preference.** We recorded total ant biomass in each bait and pooled baits such that each sampled tree was represented by four biomass data points: arboreal protein, arboreal carbohydrate, terrestrial protein, and terrestrial carbohydrate. Each biomass data point in each tree represents four collection events (2 baits x 2 collection times). To account for ants removed for identification, we calculated the average weight of a single individual of each species and used it to add the estimated biomass of each of the 1292 pinned individuals back into their sample of origin. For three species for which we could not calculate an average weight, we used the calculated average weight of a congener of similar size (S3 Table in S1 File), and we excluded one species, *Pseudomyrmex ejectus* (Smith), which was represented by a single individual across all baits. To stabilize variance, we log + 1 transformed the total biomass estimates. We then used a linear mixed model constructed with *lme4* 1.1-31 [43] to assess how ant nutrient preference shifted between habitat types and strata. We modeled the transformed biomass as a function of stratum, habitat type, nutrient, and their interactions with random intercepts of site and tree. We checked model fit with diagnostic plots and tests in *DHARMa* 0.4.6 [44] and tested significance using Type II Wald $\chi^2$ tests in *car* 3.1-1 [45]. We calculated planned simple main effect contrasts for the three-way interaction between stratum, habitat type, and nutrient with *emmeans* 1.8.4-1 [46]. We report back-transformed marginal means and their errors. We accounted for multiple comparisons using multivariate $t$ probabilities [46]. We acknowledge the potential for non-independence between baits of opposite nutrients at each bait station. We ran the analysis including bait station as a random effect; however, the previously described analysis, which pools across bait stations within a tree, presents qualitatively the same results, is a better fit for the data, and is quantitatively more conservative. Thus, we present the results of that model. To visualize community nutrient preference patterns, we calculated the biomass of every ant species for each combination of habitat type, stratum, and nutrient and plotted one bipartite network per habitat type using *bipartite* 2.18 [47,48].

Finally, we note that community-level differences in nutrient use between forest and urban sites could arise from the following non-exclusive scenarios: (a) consistent changes in the behavior of many individual species that occur in both habitat types, (b) shifts in the behavior of one or a few species that dominate the community biomass, or (c) the distinct preferences of species that were predominantly or uniquely urban. To understand how individual species' preferences shifted with urbanization (scenario a), we calculated the proportion of each species' biomass collected from arboreal protein and terrestrial carbohydrate baits within each site. We only included species which occurred in both forest and urban sites (N = 16) and fit a beta regression model constructed with *glmmTMB* 1.1.5 [49]. We modeled the proportions as a function of habitat type with a random intercept of species, checked model fit with diagnostic plots, and tested statistical significance using Type II Wald $\chi^2$ tests in *car* 3.1-1. For scenario

(b), we considered species to be biomass-dominant if they represented more than 10% of the total biomass in a given habitat type. First, we visually assessed the two bipartite networks for evidence of behavioral changes between strata and habitat types. Then to test for behavioral changes in each of the urban dominant species, we fit a linear mixed model with *lme4* to the species-specific log + 1 transformed biomass pooled in the same manner described above, i.e., each combination of nutrient and stratum in each tree. To avoid zero inflation and conflating species' absence with lack of preference, we only included biomass data from trees and strata in which the species was collected. Thus, if we only recorded a species' biomass in arboreal protein baits in a given tree, we included a zero for the arboreal carbohydrate baits, but did not include zeros for the terrestrial baits at that tree, since the species was not detected terrestrially. We modeled the transformed biomass as a function of stratum, habitat type, nutrient, and their interactions with random intercepts of site and tree and tested significance using Type II Wald $\chi^2$ tests. We calculated planned simple main effect contrasts for any significant interactions and accounted for multiple comparisons using multivariate *t* probabilities. For scenario (c), we considered two assemblages: uniquely urban species and introduced species of tropical origin. We repeated the log + 1 transformed linear mixed model with fixed effects of stratum, nutrient, and their interaction and random intercepts of site and tree to understand how the addition of these species to the urban matrix could contribute to the observed preference patterns.

## Results

After excluding missing and damaged baits, we collected 1321 (out of 1344) individual baits from 83 trees (42 forest, 41 urban) across seven urban and seven forest sites. Ants occupied 786 baits across all sites and trees and were detectable in both strata (arboreal: 300 baits, terrestrial: 486). We collected ants from 39% of baits in the forest canopy, 65% on the forest floor, 52% in the urban canopy, and 82% on the urban ground. We identified a total of 47 ant species in 17 genera and five subfamilies (S4 Table in S1 File), representing more than a quarter of the 164 species known from this region [50].

### Diversity and community composition

We observed the most species in the urban terrestrial habitat (31 species) and the fewest in the forest arboreal habitat (18 species). All habitats had a similar number of unique, non-singleton species (two species each in urban arboreal, forest arboreal, and forest terrestrial; 3 species in urban terrestrial). In urban sites, 14% of species were detected only at arboreal baits, 36% terrestrial, and 50% in both strata. In forests, 19% of species were arboreal, 31% terrestrial, and 50% in both strata. Chao2 asymptotic estimators of species richness were similar across the habitats (urban arboreal: 28.2 species, urban terrestrial: 37.1, forest arboreal: 26.7, forest terrestrial: 27.9) and shared overlapping 95% confidence intervals (Fig 2a). Other measures of diversity followed the same pattern (S2 Fig in S1 File). Based on these results, and reasonably high sample coverage in all four habitats (urban arboreal: 0.914, urban terrestrial: 0.913, forest arboreal: 0.826, forest terrestrial: 0.882), we conclude that ant diversity at baits did not differ between strata and habitat types.

Ant community composition depended on the habitat \* stratum interaction (PERMANOVA: $F_{(1,12)} = 2.62$, $R^2 = 0.047$, $p = 0.013$) and the additive effect of site (PERMANOVA: $F_{(12,12)} = 1.81$, $R^2 = 0.388$, $p = 0.001$). Because the community variances did not differ (PERMDISP2: $F_{(3,24)} = 0.546$, $p = 0.655$), we attribute the significant PERMANOVA to differences in the community centroids. All pairwise comparisons between levels of the habitat \* stratum interaction were statistically significant (Fig 2b, S5 Table in S1 File). In both forests and urban

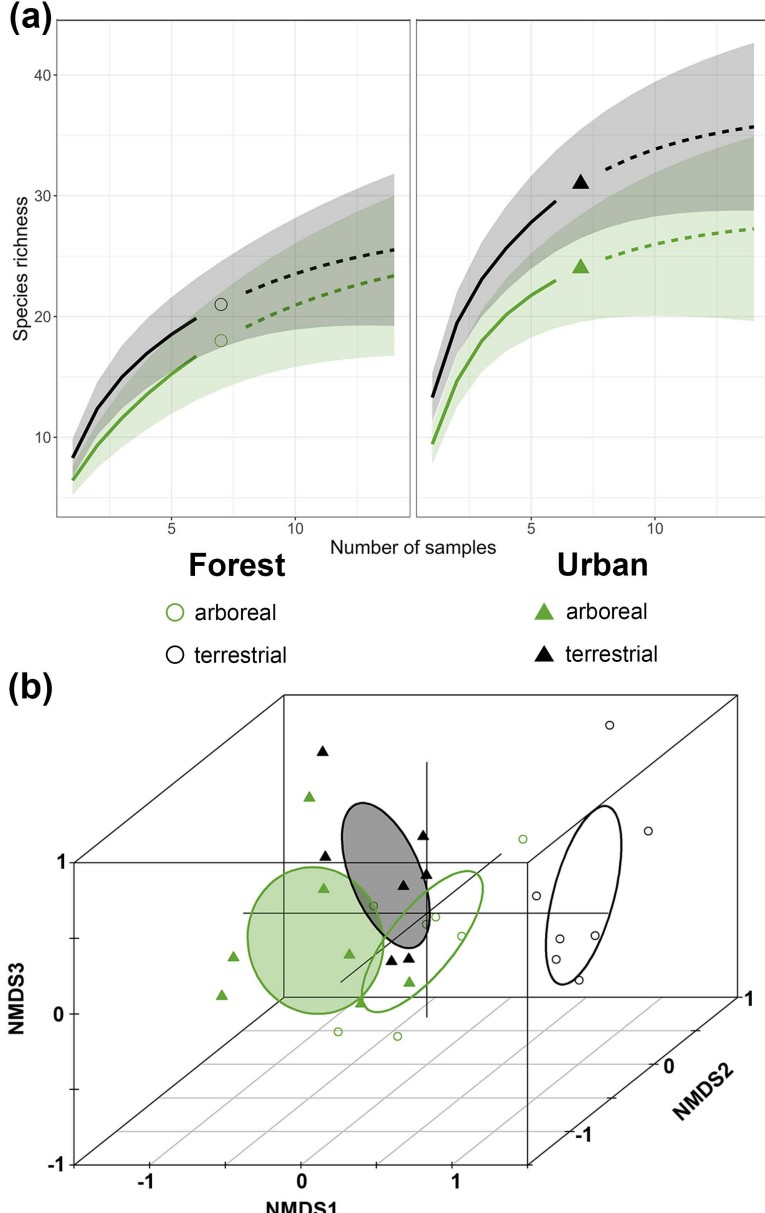

**Fig 2. Ant diversity did not differ between strata and habitats, but species turnover resulted in four distinct communities.** (a) Sample-based rarefaction (solid lines) and extrapolation curves (dashed lines) of ant species richness across vertical strata (canopy/ground) and habitat types (forest/urban). Shaded regions represent 95% confidence intervals. Observed species richness is denoted by a circle (forest) or triangle (urban). (b) Nonmetric multidimensional scaling (NMDS) plot of ant community composition across vertical strata and habitat types. Shaded (urban) and open (forest) ellipses represent 95% confidence regions of group centroids. Stress = 0.09.

areas, community differences between the canopy and ground were largely due to species turnover rather than nestedness (forest: average nestedness = 0.16, average turnover = 0.59; urban: average nestedness = 0.13, average turnover = 0.56) (S3 Fig in S1 File). Turnover was also mainly responsible for community differences between the urban and forest communities in each stratum (arboreal: average nestedness: 0.14, average turnover: 0.60; terrestrial: average nestedness: 0.10, average turnover: 0.77) (S4 Fig in S1 File).

## Nutrient preference

As expected for a temperate system, we collected significantly more ant biomass on the ground (Table 1, S5 Fig in S1 File), but 33.6% of our total ant biomass was found in the canopy, suggesting that the tree canopy still provides valuable resources for temperate ant communities. Species that foraged both in the canopy and on the ground accounted for 98% of biomass in forest and urban sites. Community-wide nutrient preferences differed among strata and habitat types. (Table 1). In forests, ants showed no nutrient preference in either stratum. In urban sites, however, ants in the canopy preferred protein (p < 0.001), while ants on the ground tended to prefer carbohydrates (p = 0.082) (Fig 3, S6 Table in S1 File). We investigated three scenarios by which these distinct, community-wide preference patterns could occur.

**Scenario a: Consistent shifts in multiple species' behaviors.** At the species level, nutrient preference patterns were consistent across habitat types within each stratum (terrestrial: $\chi^2 = 0.003$, df = 1, p = 0.958; arboreal: $\chi^2 = 1.73$, df = 1, p = 0.188), indicating that the urban preference shift was not due to a consistent, multi-species shift in nutrient use between forest and urban sites.

**Scenario b: Shifts in dominant species' behaviors.** We identified four dominant urban species (>10% of total urban biomass each). Of these, *Crematogaster ashmeadi* Mayr is responsible for the urban arboreal protein preference. *Crematogaster ashmeadi* foraged almost exclusively for arboreal protein in both habitat types (p = 0.006) (Fig 4, S7 Table in S1 File) but was absolutely and relatively more abundant in urban sites (p < 0.001) (Fig 4, S7 Table in S1 File). We collected this species from a similar number of trees in each habitat type (20 forest, 22 urban), but measured over 5.5 times more *Cr. ashmeadi* biomass in urban trees (forest: 243.5mg, urban: 1345.2mg), leading to more overall ant biomass in urban arboreal protein than urban arboreal carbohydrates, and thus contributing to the urban arboreal protein preference. The other three biomass-dominant taxa did not appear to contribute to the forest-urban preference shift. *Formica subsericea* Say preferred terrestrial carbohydrates in both habitat types (p < 0.001), and *Camponotus pennsylvanicus* (DeGeer) preferred protein in both strata and habitats(p < 0.001), but both taxa made up a similar or smaller share of community biomass in urban sites than forest sites (Fig 4, S7 Table in S1 File). *Solenopsis invicta* Buren was more abundant in the city (forest biomass: 0.6mg, urban biomass: 838.5mg), but foraged equally on carbohydrates and protein in both strata (Fig 4, S7 Table in S1 File).

**Scenario c: Addition of uniquely urban and tropical species.** The 17 species unique to urban sites significantly preferred terrestrial carbohydrates to terrestrial protein

**Table 1. Ant nutrient preferences depend on habitat and vertical stratum.**

| Predictor | $\chi^2$ | df | p(>$\chi^2$) |
|---|---|---|---|
| Stratum: *terrestrial/arboreal* | 46.8 | 1 | **<0.001** |
| Habitat: *forest/urban* | 0.7 | 1 | 0.417 |
| Nutrient: *protein/carbohydrate* | 0.6 | 1 | 0.444 |
| Stratum*Habitat | 0 | 1 | 0.891 |
| Stratum*Nutrient | 12.4 | 1 | **<0.001** |
| Habitat*Nutrient | 0.6 | 1 | 0.445 |
| Stratum*Habitat*Nutrient | 7.1 | 1 | **0.008** |

Wald Type II (marginal) $\chi^2$ tests of the linear mixed model predicting log + 1 transformed ant biomass as a function of stratum, habitat, nutrient, and their interactions. Marginal $R^2 = 0.131$, conditional $R^2 = 0.394$. Values in bold indicate statistical significance at the 0.05 level.

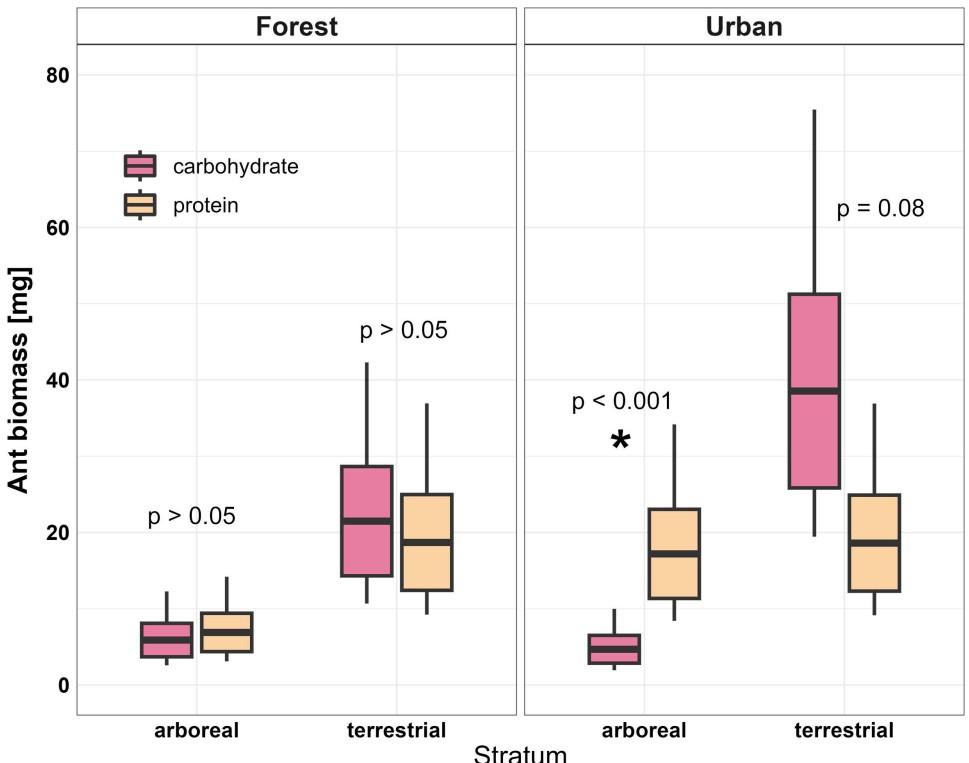

**Fig 3. Ants in temperate forest habitats showed no nutrient preferences, while ants' preferences in nearby urban habitats differed between vertical strata.** Ant nutrient preference in the canopy and on the ground of temperate forest and urban habitats. Boxes are model-predicted means ± SE of ant biomass and whiskers are 95% confidence intervals. Asterisk indicates p-value < 0.05.

(stratum*nutrient: $\chi^2 = 5.50$, df = 1, p = 0.019; terrestrial C—terrestrial P: $t_{(72.6)} = 3.76$, p = 0.0007), suggesting the observed urban terrestrial carbohydrate preference resulted from the addition of these species. Results were qualitatively similar when we considered only the 13 urban species that were native. Moreover, the three non-native tropical species (*Brachymyrmex patagonicus* Mayr, *Linepithema humile* (Mayr), and *Solenopsis invicta*) showed no nutrient preference regardless of stratum (stratum*nutrient: $\chi^2 = 0.282$, df = 1, p = 0.596; nutrient: $\chi^2 = 1.64$, df = 1, p = 0.201). Thus, our data suggest that the addition of native species to the urban habitat is the main driver of the Lasius neoniger preference (Fig 4 see *Formica biophilica* Trager, *Lasius neoniger* Emery, and *Tapinoma* cf. *sessile*; S2 Text in S1 File).

## Discussion

We assessed community patterns of ant diversity and compared nutrient preferences across two vertical strata in temperate forests and an urban metropolis. Whereas tropical studies find strong vertical stratification of ant diversity and nutrient preference [e.g., 3,5], we detected neither pattern in our North American temperate forest sites. In nearby urban trees, however, we did observe the vertically stratified nutrient preferences seen in many tropical systems, such that ants in the canopy preferred protein, while ants on the ground preferred carbohydrates [e.g., 2,5] (Fig 3).

Contrary to our predictions, ant diversity did not differ between vertical strata or habitat types. Each community (urban arboreal, urban terrestrial, forest arboreal, forest terrestrial)

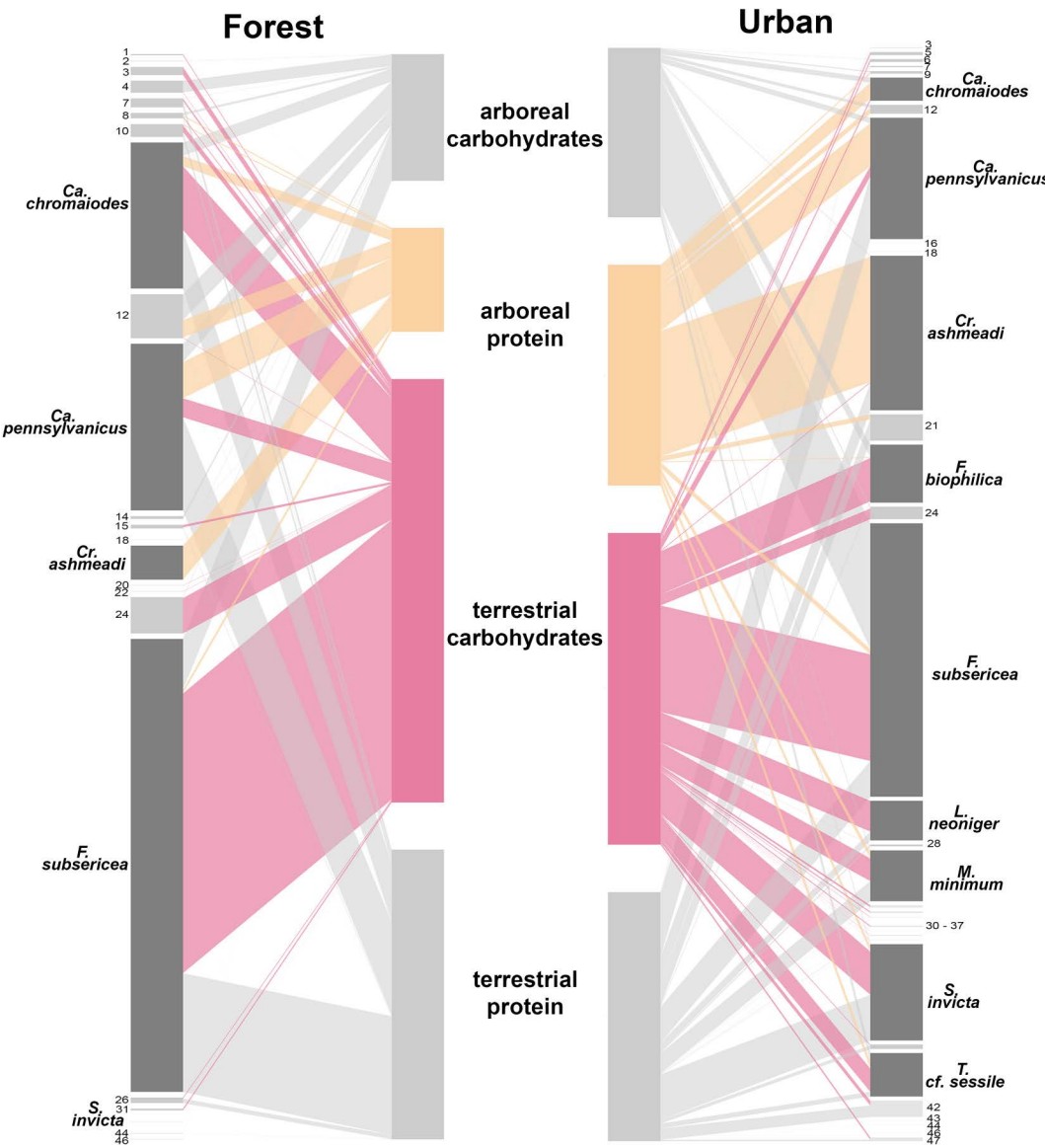

**Fig 4. Changes in native species' behaviors and the addition of uniquely urban species drove the observed preference shift between habitats.** Bipartite interaction networks of ant species nutrient preference across vertical strata (canopy/ground) and habitat types (forest/urban). Species bars (outer) represent total relative biomass of a species collected within a habitat. Nutrient bars (inner) represent total relative ant biomass collected in a nutrient within a stratum and habitat. Width of connecting lines represent total relative biomass of a species collected in a nutrient within a stratum and habitat. Nutrients for which we observed a preference shift across habitat types are highlighted (arboreal protein: tan, terrestrial carbohydrates: pink). Species that make up a large portion of community biomass and/or show a behavioral shift across habitat types are highlighted (dark gray) and named. Refer to S4 Table in S1 File for remaining species. Genus abbreviations are as follows: Ca. = *Camponotus*, Cr. = *Crematogaster*, F. = *Formica*, L. = *Lasius*, M. = *Monomorium*, S. = *Solenopsis*, T. = *Tapinoma*.

had a distinct species composition (Fig 2, S5 Table in S1 File) driven mainly by species turn-over (S3 and S4 Figs in S1 File). Surprisingly few studies make direct comparisons between urban and non-urban ant communities; our results are consistent with the few prior terrestrial studies that find no difference in alpha diversity between urban and non-urban sites [51,52], or peak diversity in suburban areas [53]. However, our diversity results should be interpreted

with caution because baiting does not sample the entire ant community [54] and is subject to effects of community dominance patterns [55].

The vertical stratification of C:N ratios at the producer level is similar in temperate and tropical forests [13], but while tropical ants often expend more foraging effort on the limiting nutrient in each stratum, the temperate forest ant communities studied here contributed equal biomass to protein- and carbohydrate-rich baits, regardless of vertical stratum (Fig 3). Similarly, Fowler, Lessard, and Sanders [12] demonstrated a lack of vertical stratification in nutrient preferences across three habitat strata (ground, shrubs, tree trunks) for the ants of Great Smoky Mountains National Park, less than 500km west of our study sites. However, due to regional or methodological differences, those communities distinctly preferred carbohydrates in every stratum.

The complete lack of nutrient preferences in our temperate forest ant communities likely reflects the ability of temperate ant species to forage flexibly in both strata. Whereas studies of tropical ant nutrient preferences found only 0 – 16% of species foraging in both strata [3,5,11], we detected 50% of forest species in both strata, and these species represented 98% of forest ant biomass. Even though canopy and ground communities were statistically distinct and harbored unique species, extensive cross-stratum foraging likely masked any influence of stratum-specific nutrient profiles at the community level. Stratum specialists may still be exposed to these limitations, but they represented only a small proportion of total forest ant biomass. Our results underscore the importance of testing ecological hypotheses in different contexts. Here, we show that a well-studied, relatively consistent trophic phenomenon in tropical forests does not translate in a temperate system.

However, in nearby urban trees, we recovered the vertically stratified nutrient preferences typical of tropical studies: ants in the canopy preferred protein, while those on the ground trended toward carbohydrates (Fig 3). Urban ants thus matched the predictions of the compensation hypothesis and countered our prediction that anthropogenic carbohydrate subsidies [24,25] would eliminate this pattern in terrestrial urban ants. It is not surprising that nutrient preferences would shift in an urban matrix. Anthropogenic food inputs, patchy resource availability, and variable food quality can affect the trophic structure, niche breadth, dietary preference, and nutrient intake of a variety of urban animals [25,26,56,57]. Still, we find it interesting that the community preferences in a novel urban habitat shifted to mimic patterns known from old-growth tropical forests. Nevertheless, it seems unlikely that the observed distribution of ants directly reflects stratum-specific nutrient limitations, because the urban ant community foraged across strata just as much as the forest community did. We found no evidence that introduced tropical ants drove differences in nutrient use between habitats. Instead, our exploratory analyses suggest that urban ants' preference for protein in the canopy was best explained by the increased urban foraging intensity of a native arboreal species, *Crematogaster ashmeadi*. This species always foraged on arboreal protein but was about 5.5 times more abundant on urban baits than forest baits. Meanwhile, urban ants' preference for carbohydrates on the ground was best explained by the presence of native species with strong carbohydrate preferences that were absent from the forest. Species that were found in both urban and forest sites did not change their preferences between habitats.

Thus, we cannot conclude, based on the distribution of ants as a whole, that nutrient availability is more steeply stratified in urban than forest sites. Instead, patterns of ant distribution consistent with the compensation hypothesis may have arisen incidentally from some combination of altered urban community composition, competition, microhabitat selection, and nutrient preferences of species within that community. For example, on the urban ground, anthropogenic carbohydrate subsidies might favor species already adapted to forage at low trophic levels, such as *F. biophilica* and *T.* cf. *sessile*, without exacerbating protein limitation

[see the economics hypothesis in [2]. In the canopy, high densities of honeydew-producing insects on stressed urban hardwoods [58,59] might support larger colonies of *Cr. ashmeadi* that still have similar stoichiometry to those in the forest [60,61]. We also acknowledge that by using complex foods (tuna and jam) as baits, other components such as water, fat, and micronutrients could have influenced ant behavior. Overall, we detected striking differences in ant biomass distribution between nutrients in urban and forest sites, but future work—such as stable isotope analyses and longer-term nutrient manipulations—are needed to parse the underlying mechanisms.

## Conclusion

We provide evidence that temperate forest communities do not adhere to nutrient preference patterns predicted by the compensation hypothesis of nutrient availability. Instead, the majority of temperate ants (50% of species representing 98% of biomass at baits) commute freely between the canopy and ground, linking green and brown food webs in ways that tropical ants do not. These findings highlight the risk of making assumptions about ecosystem-and community-level processes across biomes. They also emphasize the need for more temperate forest ecology research to consider the tree canopy, where we detected more than 1/3 of foraging ant biomass. We also show, for the first time, that urbanization in a temperate zone can alter community nutrient preferences to mirror tropical patterns, though likely due to different mechanisms. As more of our world is urbanized, we are shifting community and ecosystem ecology in novel and unexpected ways. Future work that integrates nutritional ecology, trophic relationships, and nutrient availability in urban systems should provide valuable insights into some of the far-reaching effects of an urban landscape.

## Supporting information

**S1 File. Supporting information file.** Document containing 3 supplementary texts, 7 supplementary tables, 5 supplementary figures, and references. **S1 Text**. Detailed bait design and methods. **S2 Text.** Description of *Tapinoma cf. sessile*. **S3 Text**. Analysis of variation in bait heights. **S1 Table.** Nutritional content of tuna and jam baits. **S2 Table.** Complete list of taxonomic keys used in this study. **S3 Table.** Substitute species biomass values. List of species for whom we could not calculate average individual weights and the species' values used as their substitute weights. **S4 Table.** Occurrence of ant species in arboreal and terrestrial baits across seven forest and seven urban habitats in central North Carolina, USA. **S5 Table.** Pairwise comparisons of changes in ant community composition. **S6 Table.** Contrasts of the three-way interaction between stratum, habitat, and nutrient from the model predicting ant biomass. **S7 Table**. Species specific biomass estimates and model results for four urban biomass-dominant species. **S1 Fig.** Bait height varied with habitat and tree species. **S2 Fig.** Ant diversity at baits did not significantly differ between habitats and strata. **S3 Fig.** Changes in ant communities between the ground and canopy were due to species turnover. **S4 Fig.** Changes in ant communities between the forest and urban habitats were due to species turnover. **S5 Fig.** Ant biomass was significantly greater on the ground than in the canopy.
(DOCX)

## Acknowledgments

Kirsten Keleher and Kayla Beckwith picked thousands of ants out of tuna and jam, Peter Willadsen put his Big Shot™ skills to the test, James Trager consulted on *Formica* and *Tapinoma* identifications, and Keary Dosier generously lent his drill press. Thank you to the Brandt family for private land access and genuine hospitality. Ed Corey (NC State Parks), Kim Radewicz

(NC State Parks), Zac Cowan (NC State Parks), Scott Letchworth (NC State Parks), Chris Baranski (NC Wildlife Resources Commission), Sally Thigpen (Raleigh Parks, Recreation, and Cultural Resources), and Alex Johnson (Durham Urban Forestry Division) facilitated research permissions and site access. We appreciate the residents of Raleigh and Durham who tolerated this work in their front yards.

## Author contributions

**Conceptualization:** Michelle Kirchner, Clyde Sorenson, Elsa Youngsteadt.

**Formal analysis:** Michelle Kirchner.

**Funding acquisition:** Clyde Sorenson, Elsa Youngsteadt.

**Investigation:** Michelle Kirchner, Lucie Ciccone.

**Methodology:** Michelle Kirchner, Clyde Sorenson, Elsa Youngsteadt.

**Project administration:** Michelle Kirchner.

**Visualization:** Michelle Kirchner.

**Writing – original draft:** Michelle Kirchner.

**Writing – review & editing:** Michelle Kirchner, Lucie Ciccone, Clyde Sorenson, Elsa Youngsteadt.

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
