## [Decision Letter · Decision Letter 0]

21 Jun 2024

Dear Dr. Kirchner,

Thank you for submitting your manuscript to PLOS ONE. After careful consideration, we feel that it has merit but does not fully meet PLOS ONE’s publication criteria as it currently stands. Therefore, we invite you to submit a revised version of the manuscript that addresses the points raised during the review process.

We look forward to receiving your revised manuscript.

Kind regards,

Daniel de Paiva Silva, Ph.D.

Academic Editor

PLOS ONE

Journal Requirements:

MK was supported through a Research Assistantship with North Carolina State University Department of Entomology and Plant Pathology. 

4. Please include your tables as part of your main manuscript and remove the individual files. Please note that supplementary tables (should remain/ be uploaded) as separate "supporting information" files.

Additional Editor Comments:

Dear Dr. Kirchner,

After this first review round, all three reviewers believe your manuscript deserves to be accepted to be published in PLoS One after major/minor reviews.

In general, all three reviewers enjoyed reading your research, but they believe both structural (written English) and theoretical improvements could be implemented to your study. Please do not forget to prepare a rebuttal letter to the reviewers informing about all the changes you implemented and justifying tose you did not accept. Considering the amount of improvements, I will grant you a three-month period to deliver the new version of the study. Please do no hesitate to write me in case you need more time. Otherwise, you also do not have to hesitate to resubmit earlier than my established due in cse you can.

Sincerely,

Daniel Silva

Reviewers' comments:

Reviewer's Responses to Questions

**Comments to the Author**

1. Is the manuscript technically sound, and do the data support the conclusions?

Reviewer #1: Yes

Reviewer #2: Yes

Reviewer #3: Partly

2. Has the statistical analysis been performed appropriately and rigorously?

Reviewer #1: Yes

Reviewer #2: Yes

Reviewer #3: Yes

3. Have the authors made all data underlying the findings in their manuscript fully available?

Reviewer #1: No

Reviewer #2: Yes

Reviewer #3: Yes

4. Is the manuscript presented in an intelligible fashion and written in standard English?

Reviewer #1: Yes

Reviewer #2: Yes

Reviewer #3: Yes

**Reviewer #1:**  I liked it very much because it is very well written, easy to follow, well organized in terms of data analysis, and following results and discussion sections. The figures in the main text are also very well constructed. Another positive aspect is the scenarios (and analyses) and details on results regarding the ants related to each scenario. Further comments can be found below:

- line 129: vouchers will be deposited? They should be prior to publication;

- line 136: citing Rstudio should be ‘RStudio Team (2023). RStudio: Integrated Development for R. RStudio, PBC, Boston, MA URL http://www.rstudio.com/.’

- line 144: please, explain in more detail which means the ‘additive effect of site’;

- line 171: checked diagnostic plots for the mixed model should be included in supplementary material;

- line 194: tropical species? It’s possible understand that in lines 260-261; needing clarify this at the first approach;

- line 205-208: would be important to show the average number of species by tree (and sd) to understand the diversity dimension in trees;

- line 207: please, clarify meaning of ‘urban arboreal, forest arboreal, combined with numbers [forest terrestrial: 2’]

- line 280-281: should also add that baiting does suffer of dominance effect in ant sampling;

- line 291-292: this is a central find in the study; on the other hand, would be expected because canopy structure in tropical forest is much more compartmentalized than temperate canopy (taller trees, without branches near to the ground, large biomass of epiphytic vegetation, etc); perhaps this should be added because there are huge differences between the two systems (temperate vs. tropical trees);

- line 333-334: that is strange – such information did not appear in the M&M section, but it is showed in almost last message of the study;

- Table S4: row number 34 is composed by two species names.

**Reviewer #2: ** Find below my comments and suggestions about the study. I found it very interesting and relevant. You present a very robust and well-constructed design and data analysis with the corresponding interpretations. My only concern in a few analyses is about you may be not taking into consideration the dependence of sampling from the same trap. We know that the abundance of workers (and hence biomass) may be related to factors such as proximity to the nest or other sources of food. In this sense, there is a dependency on the biomass of ants of one species in one bait with the biomass on the other bait of the same trap (see below). Perhaps I am wrong, but I suspect that considering the absolute biomass or not using the trap as a random variable may risk misleading conclusions, at least theoretically.

Detailed comments

• Line 72-74: I agree with your general statement that differences in vertical structure may translate into differences in animal preferences. However, you should be more specific about what those differences are between forest types (tropical vs temperate) to better highlight the consequential effects on animals.

• Lines 95-102: The main hypothesis and predictions are well introduced. They concern two main response variables: ant diversity and food preference, in relation to two main predictors: strata and urbanization level. Nevertheless, for the first response (diversity), you presented expectations for predictors only as independent variables, while for the second response (preference), the expectation implies an interaction between predictors. As many of the further analyses consider the interaction among predictors, I suggest mentioning the expectations about the interaction for ant diversity and possible mechanisms behind it.

• Lines 112-113: Perhaps you could describe (at least qualitatively) the general structure of the vegetation (in both forest and urban sites), specifically the tree height and vegetation density.

• Lines 113-117: As community richness may be dependent on habitat area size, you should provide the sizes of sampling areas or at least some estimation of their variability. Could this be responsible for some of the observed variation?

• Lines 121-122: I found this bait station design very interesting and promising. It may help address nutrient preferences in other habitats. However, I think you should mention (or test, if possible) how (or if) dominance relationships among ant species on bait finding may introduce bias in the results. Also, consider that this design introduces a degree of dependency of samples with different baits from the same trap, which should be considered in the data analysis.

• Line 127: Were samples from different exposure times (4h and 8h) taken as independent samples or pooled together? Also, were baits replaced after 4h removal? Otherwise, could any chemical signals affect subsequent samples? (after reading the results I understood that this data were pooled together, you should mention this here)

• Lines 131-133: Were ants from the same species weighed pooled together (and a mean value calculated) or individually?

• Lines 141-142: You should explain the purpose of removing singletons.

• Lines 152-153: Perhaps I missed it, but I couldn’t find the results of betadisper neither in the results nor in the interpretation in the discussion.

• Lines 156-158: Was there any reason for not performing the same beta partition procedure among urban and forest communities?

• Line 160: I calculated the total number of trees following the description of the sampling method, reaching 84 trees in total. Am I right? If so, what happened to the missing tree? (You explain this later, but putting 83 at this point of the text is confusing.)

• Line 166: You should report the description year along with the author for scientific names.

• Line 170-171: Did you consider including the trap as a random variable? I think it is very important to account for dependency issues among baits of the same trap and hence the comparison among strata.

• Lines 187-188: Was there a reason for not using the same package (glmmTMB) in the previous GLMMs? If possible, use the same; the results should be equal or very similar, but you’ll maintain consistency in estimation.

• Lines 190-192: I think you should better justify your criteria to define dominant species. While total biomass alone may be a relevant factor, it may underestimate the importance of small but aggressive species or even those with high recruiting capacity.

• Lines 192-193: This network approach seems interesting, but I found it confusing when trying to understand how it helps to clarify scenario b. It appears somewhat subjective (here you state that you visually assess the networks, but you later calculated some metrics) and fails to consider the particularity of your sampling design. Specifically, it suggests that all biomass measures (ground protein, ground carb, arboreal protein, and arboreal carb) are independent and directly comparable. Are they?

• Lines 200-201: Perhaps you can also report the number (or even better the proportion) of baits occupied by ants in the urban and forest sites.

• Lines 220-221: Was there a reason to use a 3D NMDS instead of a 2D NMDS to represent composition differences among habitat types? A 2D representation would be more easily interpreted, and I would prefer it as long as the stress remains somewhat low.

• Lines 240-241: I disagree with this interpretation. Connectance values seem to be very close to say safely that they are indeed different. You may provide a more objective evaluation, perhaps a null model approach.

• Lines 241-254: I think you may support these statements about main dominant species with some estimation of effect size for individual species, such as estimated standardized coefficients from mixed models.

• Lines 249-252: You should report biomass estimates from habitats for both C. pennsylvanicus and F. subsericea. Perhaps it could be even better to include a table with biomass by species in the appendix.

• Lines 295-299: Your conclusion about differences in ant fauna composition is based on presence/absence data. If you compare the communities based on abundance or biomass-weighted matrices, do you think that it may better reflect the role of cross-habitat dominant species on community structure?

• Line 341: Do you mean the majority of species or individuals?

Finally, during the results, you highlight the different patterns derived from tropical invasive species. However, this topic is not treated in more depth in the discussion.

**Reviewer #3: ** In this manuscript, Michelle Kirchner, Elsa Youngsteadt and collaborators studied the effects of urbanization on a temperate forest ant community, regarding both vertical stratification (arboreal vs. ground) and macronutrients preference (carbohydrates vs. proteins). For doing this they compared seven sites in two US cities (Raleigh and Durham; North Carolina State) against seven sites in protected, secondary mixed hardwood forests in state parks, game lands, or private properties. At each site, they selected three trees of two species each (Quercus alba L. and Acer rubrum L.) and installed baits in the canopy and on the ground (two baits per strata at each tree). They present a novel bait station, consisting of a pine dowel with two styrene tubes attached to both ends, each containing a distinct bait type (either a carbohydrate-rich food, strawberry preserves, or a protein-rich food, tuna) and bearing drilled entry holes for guaranteeing ant access to the baits (see Supplementary material, Text S1, for more details). Their predictions were that (1) ants foraging in the canopy will prefer protein, while those on the ground will prefer carbohydrates; and (2) urbanization would eliminate nutrient preferences on the ground, but nutrient preferences in the canopy would remain unchanged.

Putting together, their results show that:

1) Ant community in both strata (ground and canopy) moderately overlapped in terms of species diversity and composition, both in urban and forest sites, what differs from tropical studies (such as Yanoviak and Kaspari 2000).

2) Although there was no nutrient*strata variation in ant preference regarding the forest habitat, the authors found that in the urban habitat, ants foraging on the canopy significantly preferred protein baits, whereas carbohydrates were almost significantly preferred in the ground. What was found for the urban habitat matches patterns described by tropical studies.

In spite of its merits, in its present form the manuscript needs to pass through moderate to major revisions in order to bring out the best of what has been discovered. My two major points of concern are the following:

1) Canopy height variation (see Page 6, Lines 127-129: “At each tree, we placed four bait stations, two in the canopy (mean height: 14.6m, range: 2.1–31.1m) and two on the ground (5m N and S of the base of the tree) (Fig. 1c) between 9:00 and 11:30am)”. – There is a huge difference between minimum (2.1 m) and maximum (31.1 m) values, and we are not informed about the standard deviation of their sampling heights. Note that if SD is high, we are probably dealing with heights that are very different to compare. While I do expect that ants foraging at 3 m height could easily forage on the ground, I do not expect that ants foraging at 25 m or more would come to the ground to forage. I am also asking myself if there is a specific bias in tree height related to the habitat: for example, that urban trees present smaller heights than forest tress. Therefore, authors should inform about sampling height for each habitat separately (mean and standard deviation). Moreover, they should discuss in which extent this variation could have affected their results (for example in the fourth paragraph of the discussion). Comparing with other studies cited by the authors, we are able to see that they present a smaller canopy height variance:

a. Yanoviak and Kaspari (2000, Oikos): canopy sampling occurred between 17 and 35 m above the ground;

b. Antoniazzi et al. (2020, Sociobiology): baits were installed at 10-15 m as high in the canopy as possible (10-15 m above the ground, depending on tree height).

2) In the discussion, I missed any mention to the three tropical invasive ants that were found in this study (Brachymyrmex patagonicus Mayr, Linepithema humile (Mayr), and Solenopsis invicta). Even though, in the results, Solenopis invicta is mentioned as being more abundant in the city (forest biomass: 0.6mg, urban biomass: 838.5mg) than in the forest, and forages equally on carbohydrates and protein in both strata (see page 12, lines 259-260). Based on Table S4, we can see that these three species seem to be slightly more common in the terrestrial than in the arboreal community. Does this fat present any consequence to the patterns discussed for urban areas?

MINOR COMMENTS

WRITING: Despite the fact that I am not a native English speaker, and that the authors are native speakers, I have made a few suggestions about words changing as well as sentences organization. These suggestions are included directly in the revised manuscript attached. Note that they are not mandatory.

KEYWORDS: I have made some suggestions regarding keywords choice directly in the revised manuscript.

ABOUT THE TREE-SPECIES USED: two tree species (white oak, Quercus alba L.; and red maple, Acer rubrum L.) were used with equal numbers (three trees per site) in both habitats, thus potentially eliminating differences in ant attendance that could emerge from plant species. However, the authors did not mention if both species commonly present ant-attractants, either sap-feeders hemipterans of the family Membracidae (Hemiptera) or extra-floral nectaries. Moreover, even though both are common species in the US, it is important to mention their family names, since Plos One audience is worldwide.

I also missed any mention in the results to the identity of tree species: did this fact affect ant composition and diversity?

VOUCHERS DEPOSITION: in page 6 (lines 129-130), as well as on the Appendix with supporting information (see “Text S2: Description of Tapinoma cf. sessile”), the authors state that “Vouchers will be deposited in the North Carolina State University Insect Museum.” Normally, when research is published, vouchers are already deposited in a collection/museum and the authors mention specimens IDs for enabling readers finding the deposited specimens.

RESULTS: Page 9 (lines 199-201) - I would suggest the authors to include the total number of baiting possibilities. Following my calculations, there were 1344 baiting possibilities and having retrieved a total of 1321 is very expressive.

LITERATURE USAGE: I noticed that the authors included many relevant and new references (from 2020 ahead). Nonetheless, I would like to advise the authors to search for the following recent works, which may be interesting to include:

- Zuanon, L.A.; Leão, R.E.O.S.; Quero, A.; Neves, K.C.; Vasconcelos, H.L. Nutrient Supplementation to Arboreal Ants: Effects on Trophic Position, Thermal Tolerance, Community Structure and the Interaction with the Host-Tree. Diversity 2023, 15, 786.

- Camarota, F., Vasconcelos, H. L., Koch, E. B., & Powell, S. (2018). Discovery and defense define the social foraging strategy of Neotropical arboreal ants. Behavioral Ecology and Sociobiology, 72, 1-11.

FIGURE CAPTIONS: figure captions could not be corrected directly on the Word file. So, here are some observations:

Figure 1 – Instead of “cartoon” use the more adequate term “illustration”.

Figure 2 – I would rather suggest the authors to show the two rarefaction curves of each habitat type (forest and urban) together in a unique figure, making it easier to compare the species richness among the four combination (strata and habitat). Note that when this figure is referred in the manuscript (see page 10, lines 214-216), the four curves are compared.

Figure 4 – Please, use the more informative term “Bipartite Interaction networks” instead of “network” alone.

Figure 4 (lines 28 -31): “Nutrients for which we observed a preference shift across habitat types are highlighted. Species that make up a large portion of community biomass and/or show a behavioral shift across habitat types are highlighted and named.” – I was not able to perceive how nutrientes were highlighted in the networks, I only noted that some ant species were highlighted.

REFERENCES FORMATTING:

- The authors did not follow Plos One guidelines for references, which state the following “References are listed at the end of the manuscript and numbered in the order that they appear in the text. In the text, cite the reference number in square brackets (e.g., “We used the techniques developed by our colleagues [19] to analyze the data”). PLOS uses the numbered citation (citation-sequence) method and first six authors, et al.” . Here, the authors used names and year for citing references throughout the text.

- Moreover, the reference list is, neither numbered (as mentioned above), nor formatted as indicated in the guidelines “PLOS uses the reference style outlined by the International Committee of Medical Journal Editors (ICMJE), also referred to as the “Vancouver” style”.

- In this sense, authors should format the references of the entire manuscript.

- Please, also note that species names in the titles of some references are not italicized.

SUPPLEMENTARY MATERIAL:

- Table S4 shows all species collected: They are sorted alphabetically and no mention to their subfamilies is made. I would recommend the authors to include subfamily names and organize species by subfamily and then inside each subfamily, alphabetically. I would also suggest the authors to show in this table the effective number of trees (or baits) harboring each ant species. At the present version, authors show only the number of sites (maximum is seven for each habitat) where each ant was found.

- The list of supporting materials (text, Tables and figures) is not shown at the end of the manuscript, as stated in Plos One guidelines: “List supporting information captions at the end of the manuscript file. Do not submit captions in a separate file.”

- Authors should also avoid mentioning “Appendix S1” everytime on the text that they want to refer to any supplementary material. It is better to only mention the specific code/number of this material. Instead of “(Appendix S1: Text S2)”, just mention “(Text S2)”.

- I would like to suggest authors to improve almost all captions for the supplementary material in order to give the potential readers more details of the study, which otherwise are found only in the manuscript file. By finding more information in the captions, readers will understand/remember study details better.

Note that some other small comments regarding methodology, references, and others were also included directly on the revised manuscript file.

**Do you want your identity to be public for this peer review?** For information about this choice, including consent withdrawal, please see our Privacy Policy

Reviewer #1: No

Reviewer #2: **Yes: ** Sebastian F. Sendoya

Reviewer #3: No

---

## [Author Response · Author response to Decision Letter 1]

17 Sep 2024

Thank you for the positive feedback and the opportunity to revise our manuscript! We address all reviewers’ comments in detail in the attached response document; line numbers in the response refer to the tracked changes version of the manuscript.

---

## [Decision Letter · Decision Letter 1]

14 Nov 2024

Dear Dr. Kirchner,

Thank you for submitting your manuscript to PLOS ONE. After careful consideration, we feel that it has merit but does not fully meet PLOS ONE’s publication criteria as it currently stands. Therefore, we invite you to submit a revised version of the manuscript that addresses the points raised during the review process.

We look forward to receiving your revised manuscript.

Kind regards,

Daniel de Paiva Silva, Ph.D.

Academic Editor

PLOS ONE

**Journal Requirements:**

**Additional Editor Comments:**

Dear Dr. Kirchner,

After a new review round, both reviewers considered your manuscript has improved significantly. Still, some minor improvements are still needed. Therefore, I will grant you a one-month period for you to deliver the corrected version of your manuscript. Please do not hesitate in case you are able to submit it earlier than proposed due. Congratulations on your hard work.

Sincerely,

Daniel Silva

Reviewers' comments:

Reviewer's Responses to Questions

**Comments to the Author**

Reviewer #2: All comments have been addressed

Reviewer #3: All comments have been addressed

2. Is the manuscript technically sound, and do the data support the conclusions?

Reviewer #2: (No Response)

Reviewer #3: Yes

3. Has the statistical analysis been performed appropriately and rigorously?

Reviewer #2: (No Response)

Reviewer #3: Yes

4. Have the authors made all data underlying the findings in their manuscript fully available?

Reviewer #2: Yes

Reviewer #3: Yes

5. Is the manuscript presented in an intelligible fashion and written in standard English?

Reviewer #2: Yes

Reviewer #3: Yes

**Reviewer #2:**  Dear Authors,

After carefully considering your explanations and the revisions made to the manuscript, I am pleased with the adjustments, particularly regarding the concerns I had about the statistical models used in the previous version. I find your responses satisfactory and appreciate the effort you put into addressing these points.

It may still be worth mentioning briefly—perhaps even in the supplementary material—how you handled the potential issue of non-independence of baits, as you explained well in your response. Including this could provide helpful context for future readers but it is not a condition to aceptance.

In light of these improvements, I am happy to recommend the acceptance of your manuscript.

Best regards,

**Reviewer #3:**  In this new version of their manuscript, Michelle Kirchner, Elsa Youngsteadt and collaborators apparently addressed and solved most of the points raised by me and the other two reviewers.

In the previous version, my principal concern was on how bait height was distributed between urban and forest sites, as well as if bait height was too low for not allowing stratification of ants between canopy and ground strata. This concern was addressed and clarified by the authors, with some inserts in the manuscript itself as well as the addition of some supporting material (Text S3, andd Figure S1).

Manuscript formatting was also thoroughly revised by authors. However, I feel that figure captions and tables are still not entirely correct. According to the model given by Plos One, they should be inserted on the text separated by a blank line

I attached a revised version of the manuscript with very few (and small) corrections.

**Do you want your identity to be public for this peer review?** For information about this choice, including consent withdrawal, please see our Privacy Policy

Reviewer #2: No

Reviewer #3: No

---

## [Author Response · Author response to Decision Letter 2]

27 Dec 2024

We thank the reviewers for their time, consideration, and helpful comments which have improved our manuscript! We provide a response document detailing our responses to specific comments. In short, we added a statement to justify our choice of pooled model in the methods section, and we adjusted some wording choices in the text.

---

## [Decision Letter · Decision Letter 2]

18 Feb 2025

Urbanization strengthens vertical stratification of ant nutrient preferences in a temperate forest ecosystem

PONE-D-24-15366R2

Dear Dr. Kirchner,

We’re pleased to inform you that your manuscript has been judged scientifically suitable for publication and will be formally accepted for publication once it meets all outstanding technical requirements.

Kind regards,

Daniel de Paiva Silva, Ph.D.

Academic Editor

PLOS ONE

Additional Editor Comments (optional):

Dear Dr. Kirchner,

I am pleased to inform you that your manuscript has been formally accepted for publication in PLoS One! Congratulations!

Sincerely,

Daniel Silva

Reviewers' comments:

Reviewer's Responses to Questions

**Comments to the Author**

Reviewer #2: (No Response)

Reviewer #3: All comments have been addressed

2. Is the manuscript technically sound, and do the data support the conclusions?

Reviewer #2: Yes

Reviewer #3: Yes

3. Has the statistical analysis been performed appropriately and rigorously?

Reviewer #2: Yes

Reviewer #3: Yes

4. Have the authors made all data underlying the findings in their manuscript fully available?

Reviewer #2: Yes

Reviewer #3: Yes

5. Is the manuscript presented in an intelligible fashion and written in standard English?

Reviewer #2: Yes

Reviewer #3: Yes

Reviewer #2: (No Response)

Reviewer #3: Dear Authors,

After reading the new version of your manuscript, I am glad to acknowledge that the changes suggested by the second round of reviewing were satisfactorily fulfilled. Therefore, I am happy to recommend the acceptance of your manuscript.

Best regards

**Do you want your identity to be public for this peer review?** For information about this choice, including consent withdrawal, please see our Privacy Policy

Reviewer #2: No

Reviewer #3: No

---

## [Editor Report · Acceptance letter]

PONE-D-24-15366R2

PLOS ONE

Dear Dr. Kirchner,

I'm pleased to inform you that your manuscript has been deemed suitable for publication in PLOS ONE. Congratulations! Your manuscript is now being handed over to our production team.

Kind regards,

on behalf of

Dr. Daniel de Paiva Silva

Academic Editor

PLOS ONE